# Diagnostic and Prognostic Deep Learning Applications for Histological Assessment of Cutaneous Melanoma

**DOI:** 10.3390/cancers14246231

**Published:** 2022-12-17

**Authors:** Sydney R. Grant, Tom W. Andrew, Eileen V. Alvarez, Wendy J. Huss, Gyorgy Paragh

**Affiliations:** 1Department of Dermatology, Roswell Park Comprehensive Cancer Center, Buffalo, NY 14263, USA; 2Department of Cell Stress Biology, Roswell Park Comprehensive Cancer Center, Buffalo, NY 14263, USA; 3Translational and Clinical Research Institute, Newcastle University, Newcastle-Upon-Tyne NE2 4HH, UK; 4Department of Pathology and Anatomical Sciences, University at Buffalo, Buffalo, NY 14260, USA

**Keywords:** melanoma, deep learning, dermatopathology

## Abstract

**Simple Summary:**

Melanoma is one of the most common malignancies in the United States. For the diagnosis of melanoma, histology images are examined by a trained pathologist. While this is the current gold standard for cancer diagnosis, this process requires substantial time and work and at a considerable cost. Moreover, histological diagnosis also adds diagnostic variability. Artificial intelligence is a valuable tool to aid this process. It can detect small image features that are unrecognizable to the human eye and improve diagnostic accuracy and prognostic classification. Here, we comprehensively review recent studies on the application of artificial intelligence for diagnosing and assessing the prognosis of melanoma based on pathology images.

**Abstract:**

Melanoma is among the most devastating human malignancies. Accurate diagnosis and prognosis are essential to offer optimal treatment. Histopathology is the gold standard for establishing melanoma diagnosis and prognostic features. However, discrepancies often exist between pathologists, and analysis is costly and time-consuming. Deep-learning algorithms are deployed to improve melanoma diagnosis and prognostication from histological images of melanoma. In recent years, the development of these machine-learning tools has accelerated, and machine learning is poised to become a clinical tool to aid melanoma histology. Nevertheless, a review of the advances in machine learning in melanoma histology was lacking. We performed a comprehensive literature search to provide a complete overview of the recent advances in machine learning in the assessment of melanoma based on hematoxylin eosin digital pathology images. In our work, we review 37 recent publications, compare the methods and performance of the reviewed studies, and highlight the variety of promising machine-learning applications in melanoma histology.

## 1. Introduction

Invasive melanoma is currently the fifth most common cancer diagnosis in the United States, with an estimated 106,000 new cases in 2022 [1]. The melanoma incidence rate has been steadily increasing in recent decades, averaging a 1.2% yearly increase from 2010 to 2019 [1]. Based on the increasing rates of melanoma worldwide, new melanoma cases are projected to increase by 50% by 2040 [2]. Although exposure to UV light has been shown to correlate with melanoma risk, the number of melanocytic nevi, family history, and genetic susceptibility are the most influential risk factors associated with the disease [3,4].

Cutaneous melanomas are diagnosed during skin examination. Clinically concerning lesions are identified based on features of asymmetry, border irregularity, color variation, increased diameter, and history of change (evolution) or frequently by dermoscopic features to improve diagnostic accuracy [5]. Concerning lesions are removed by excisional or shave biopsy or very large lesions are partially biopsied in areas most concerning for invasion for histological confirmation of diagnosis. The diagnosis of melanoma is confirmed using histological analysis of skin biopsies. Histopathology revealing an increased number of atypical melanocytes in the epidermis or dermis growing in a disorderly fashion may lead to a cancer diagnosis; however, histological diagnosis is not always clear-cut [5]. Histological analysis of melanoma draws diagnostic and also prognostic conclusions about the disease. In addition to determining histological types of melanoma, pathologists also make observations that bear prognostic significance. The predictive histological features include the presence of ulceration, mitotic rate, lymphovascular invasion, microsatellites, neurotropism, and tumor-infiltrating lymphocytes [6,7,8]. Despite many years of using these methods as the standard of care, there is still much room for improvement [9,10]. In many cases, interobserver variability between pathologists is high [10,11]. Additionally, these processes require substantial time and effort from trained pathologists. Analysis of clinical data with the use of artificial intelligence has been shown to increase the accuracy of patient diagnosis and prognosis [11,12] and has the long-term potential to lessen the current burden of analysis on pathologists.

Machine learning (ML) is a widely popular subdomain of artificial intelligence which uses computer systems to learn data-analysis tasks and improve output performance without explicit human instructions. ML algorithms use statistical analysis to recognize patterns within large datasets and make inferences on the dataset output [13]. A model is developed using a training dataset to identify features associated with certain output variables. The model may then be tested and finetuned using a test dataset that the model has never seen before. The model’s performance may then be assessed based on the sensitivity, specificity, and accuracy of predicting the correct output variables in the test dataset [14]. As ML models’ performance improves, the implementation of these technologies in clinical research has increased. One example of this includes the use of ML models to predict the length of operating room surgical cases, improving the overall accuracy and speed of prediction [15]. Although we are not yet at a stage to allow ML technologies to make clinical decisions independently, they may be used to aid physicians in practice [16]. Deep learning represents a subset of ML in which multiple processing layers of artificial neural networks independently extract features from training datasets. Deep learning has been shown to be a powerful tool for analyzing high-dimension datasets and detecting features unrecognizable to the human eye [17].

Deep learning applications have been developing to aid medical diagnosis and prognosis in recent years [17]. Deep learning models allow for the potential recognition of patterns in images and other forms of data hidden in plain sight of the human eye. The discovery of these new features allows for a possible avenue to use deep learning in conjunction with routine physician decisions to allow for more effective clinical decision-making: artificial-intelligence-augmented decision-making. Deep learning has been applied using numerous data types, including medical imaging, genomic, transcriptomic, and unique clinicopathologic features. Several of these require additional sample collection or time for imaging or questionnaires and are thus hard to integrate seamlessly into the current clinical workflow. However, histological images are already the gold standard for diagnosing most cancers, including melanoma. Additionally, histological images contain more pixels than radiological images and retain invaluable information regarding different cell types, morphology, and spatial arrangement, providing a powerful tool for novel biomarker discovery [18,19]. Here, we review the evolving new field of ML in melanoma histology. Our analysis highlights the findings from 37 studies on deep learning applications for the diagnosis and prognosis of melanoma based on hematoxylin-and-eosin-stained images. We summarize each analysis including the types of deep learning models used, dataset parameters, and model performance and provide a comprehensive overview of the status of deep learning in melanoma histology.

## 2. Materials and Methods

### 2.1. Identification of Research Articles

A comprehensive literature search was performed on 21 March 2022. We searched for the terms “melanoma” OR “skin cancer” AND “deep learning” OR “machine learning” OR “artificial intelligence” AND “histology” OR “whole slide images” OR “hematoxylin and eosin” in NCBI PubMed and on Google Scholar and SCOPUS. A total of 31,486 publications were identified with the search terms. After removing duplicates and reviewing the publications, 37 were found to contain primary research findings and descriptions of ML projects; thus, these publications were included in our study (Figure 1). The publications were categorized based on whether they aimed to identify diagnostic (*n* = 25) or prognostic features (*n* = 12) (Figure 2A). A thorough review of the publications followed to assess findings and provide a comprehensive overview of ML in melanoma histopathology.

### 2.2. Data Extraction of Research Articles

Each article was thoroughly reviewed and the main findings were recorded. Additional parameters extracted from each article included the date of publication, size of the dataset, type of machine-learning algorithms used, model application, and reported model performance.

### 2.3. Creation of Figures

All computational analysis and figures for this review were performed in R version 4.1.1 [20]. Packages utilized for data visualization include ggplot [21] and ggpubr [22]. Additional figures were generated using Biorender (https://biorender.com/, accessed on 6 October 2022).

## 3. Results

With improved computational power and image and data acquisition and handling techniques, the interest in ML for biomedical research has been increasing over recent years [14]. This trend was found to hold true based on our literature search of deep learning applications on melanoma histology. A total of 86% of the studies we found were published since 2019, i.e., within the past 3 years (Figure 2B).

A sufficient dataset size is essential for building successful and robust deep-learning models. The size of datasets within these studies ranged from 9 to 18,607 images, with a median size of 324 images (Figure 2C). The majority of cases utilized whole slide images (WSIs), while others with limited datasets split WSIs into multiple smaller images. Several studies with limited access to image databases performed validation studies using either cross-validation or split test and training datasets. However, numerous other studies validated their trained model using imaging data collected at other institutions, increasing the reliability of the performance of their reported model.

Convoluted neural network (CNN) deep-learning models were used in 67.6% of reviewed studies, more than any other type of learning model. Other groups used different neural networks, including artificial neural networks and deep-learning networks. Few studies included different ML models in their approach, including multi-support vector machines and random forest.

The beginning phases of developed algorithms followed a similar workflow for many published models. Often beginning with whole-slide images, these images were split into many small tiles. From these tiles, algorithms extracted cellular and spatial features. To draw a final conclusion on the image as a whole, the majority decision based on all tiles was used (Figure 3).

### 3.1. Diagnostic Applications

The current standard risk assessment of pigmented lesions begins with macroscopic and dermoscopic examination by a physician. Physicians look for a number of known risk factors for melanoma, including irregular borders, asymmetry, color, diameter, change in lesion over time, and comparison of the lesion to the patient’s other nevus [23]. Visual examination and dermoscopy allow physicians great accuracy in differentiation between melanoma and nevi. ML algorithms have also shown great accuracy for melanoma vs. nevi differentiation based on clinical images [24]. However, tissue biopsy is essential to achieve a formal diagnosis. Most deep-learning diagnostic applications for histological images are for the differentiation between melanoma and nevi [25,26,27,28,29,30,31,32,33]. However, multiple studies show applications in the differentiation between melanoma, nevi, and normal skin [34,35]; and differentiation between melanoma and nonmelanoma skin cancers [36,37,38]. Several studies showed deep-learning applications for the segmentation of whole tumor regions [39,40,41,42] or individual diagnostic markers such as mitotic cells [43,44], melanocytes [45,46], and melanocytic nests [47]. Several of these models were compared against the diagnostic accuracy of trained histopathologists, showing improved performance [25,27,29,31] (Table 1).

The arrangement and location of melanocytic cells are essential factors for pathologists to consider when assessing the disease status of WSIs. However, visually, cells of melanocytic origin can be difficult to differentiate from surrounding keratinocytes, even to the trained eye. Multiple groups have developed programs to identify proliferative melanocytes, aiding both the discovery of melanocytes and information on the overall melanocyte growth patterns. Liu et al. developed a model to segment melanocytic proliferations. Using sparse annotations generated by the pathologist, this pipeline finetunes the segmented regions using a CNN model on tiles WSI regions with an overall accuracy of 92.7% [46]. Kucharski et al. used a convolutional autoencoder neural network architecture to detect melanocytic nests. Slides were split into tiles, where individual tiles were classified as part or not part of a nest, eventually allowing for the segmentation of the whole nests [47]. Andres et al. used random-forest classification to classify individual tiles as tumor regions based on color components and cell density. Individual cell nuclei are detected, and the probability of each nuclei pixel being a part of a mitotic nucleus is calculated, resulting in an overall prediction of whether a cell is in mitosis. They found a significant correlation between the number of mitoses detected by their program and the number of Ki67-positive cells seen in Ki67-stained tissue slides, and their model achieved 83% accuracy for the correct prediction of mitotic cells [43].

Wang et al. tested the efficacy of multiple CNN pre-trained models for the prediction of malignancy of each slide tile. Tile prediction was used to generate a heatmap, from which additional features were extracted and used in a random forest algorithm to classify WSIs. Final predictions of the model based on validation datasets were compared to those of seven pathologists. The model outperformed human pathologists, achieving an accuracy of 98.2% [29]. Xie et al. additionally used a Grad-CAM method to reveal the logic behind the CNN and understand the impact of specific areas in the model. The use of the Grad-CAM method and feature heatmaps revealed similarities between this group’s model and accepted pathological features, ultimately leading to an overall accuracy of 93.3% [30].

Multiple publications show the efficacy of multi-class support vector machine models for the classification of skin WSI samples as melanoma, nevi, or normal skin [34,35]. Lu et al. first developed a pipeline for cell segmentation and feature extraction. This pipeline first segments keratinocytes and melanocytes in the epidermis, afterwards constructing spatial distribution and morphological features. The final model based on the most critical distribution and morphological features achieved a classification accuracy of 90% [35]. Xu et al. later expanded the model to first segment the epidermis and dermis from these images and analyzed epidermal and dermal features in parallel. This model observed similar epidermal features while performing dermal analysis focusing on textural and cytological features in those regions, achieving an improved accuracy of 95% [34].

Spitzoid melanocytic tumors are a subset of melanocytic lesions that are particularly challenging to diagnose. Therefore, there is an acute need for improved diagnostic measures for these tumors. Using a weakly supervised CNN-based method, Amor et al. created a pipeline to identify tiles of tumor regions and then classify WSIs based on the output tiles. This group’s model for ROI extraction achieved an accuracy of 92.31% and a classification model accuracy of 80% [28].

Sankarapandian et al. further expanded the utility of WSIs for melanoma diagnosis by creating an algorithm to diagnose and classify sub-types. WSIs of nonmelanoma and melanocytic lesions of varying disease classification first undergo quality control, followed by feature extraction and hierarchical clustering. Initial clustering led to a binary classification of nonmelanoma vs. melanocytic images followed by further classification of melanoma as “high risk” (melanoma) “intermediate risk” (melanoma in situ or severe dysplasia), or “rest” consisting of nonmelanoma skin cancers, nevus, or mild-to-moderate dysplasia. On their two independent validation datasets, their model achieved an AUC of 0.95 and 0.82 [38].

Differentiation between melanoma and nonmelanoma skin cancers is typically performed by visual examination. Melanoma commonly appears as a darkly pigmented lesion, while basal and squamous cell carcinomas can display various visual characteristics, including lesion scaling, erythema, and hyperkeratosis [48]. Ianni et al. used a dataset of over 15,000 WSI acquired from various institutions to ensure the reproducibility of their developed model. The acquired images were diagnosed as either basaloid, squamous, melanocytic, or with no visible pathology or conclusive diagnosis. This group utilized multiple CNN models, each serving a unique purpose in their diagnostic pipeline. By testing the model’s accuracy on images of three different labs, the model achieved an overall accuracy ranging from 90 to 98% in the prediction of the correct skin cancer sub-type [36].

The plethora of algorithms shown to diagnose nevi and melanoma accurately on histology of biopsied samples indicate great promise for the future of automatic diagnoses using deep-learning technologies. However, there is still much progress that could be made in this field. Further advances in the field may later allow for deciphering more specific melanoma traits, including tumor melanoma subtypes and high-risk features.

**Table 1 cancers-14-06231-t001:** Diagnostic Applications.

Author	Dataset Size (# of Images)	Dataset Type	Model Type	Model Application	Reported Model Performance	Main Findings
Hekler et al. [25] (2019)	Train = 595 Test = 100	H&E-stained images	CNN	Melanoma vs. nevi	Sensitivity = 76% Specificity = 60% Accuracy = 68%	Classification algorithm differentiated between melanoma and nevi H&E slides. In comparison to 11 histopathologists, the algorithm performed significantly better on the test dataset (*p* = 0.016).
Hohn et al. [26] (2021)	Train = 232 Test = 132 Validation = 67	H&E-stained images; clinical information	CNN	Melanoma vs. nevi	mean AUROC = 92.30% mean balanced accuracy = 83.17%	Classification model to diagnose melanoma and nevi whole-slide images. This group compared the efficacy of a CNN model alone to the CNN model assisted with commonly used patient data in multiple fusion model settings. They found that the independent CNN model achieved the highest performance.
Brinker et al. [27] (2021)	100	H&E-stained images	CNN	Melanoma vs. nevi	Sensitivity = 94% Specificity = 90% Accuracy = 92% AUC = 0.97 (annotated)	Transfer learning of pre-trained CNN model diagnosed melanoma and nevi whole-slide images with and without annotation of lesion region. The success of the trained model was compared to image classification performed by 18 expert pathologists. The model tested using annotated images outperformed the overall accuracy of pathologists.
Ianni et al. [36] (2020)	Train = 5070 Test = 13,537	H&E-stained images	CNN	basaloid vs. squamous vs. melanocytic vs. other	Accuracy = 90–98%	Diagnosis of either basaloid, squamous, melanocytic, or with no visible pathology or conclusive diagnosis. Multiple CNN models each served a unique purpose in their diagnostic pipeline. The accuracy of the model was tested on images from three different labs.
Amor et al. [28] (2021)	54	H&E-stained images	CNN	spitzoid melanoma vs. nevi	accuracy = 0.9231, 0.8	Computational pipeline consisted of ROI extraction to identify tiles of tumor regions followed by a classification model to diagnose WSIs.
Xu et al. [34] (2018)	66	H&E-stained images	mSVM	Melanoma vs. nevi vs. normal skin	95% classification accuracy	Model classified skin WSI samples as melanoma, nevi, or normal skin. The model first segments the epidermis and dermis, analyzes epidermal and dermal features in parallel, and finally classifies the image into one of three classes.
Lu et al. [35] (2015)	66	H&E-stained images	mSVM	Melanoma vs. nevi vs. normal skin	90% classification accuracy	This pipeline first segments keratinocytes and melanocytes in the epidermis, then constructs spatial distribution and morphological features for WSI diagnosis of melanoma, nevi, and normal skin.
Wang et al. [29] (2019)	155	H&E-stained images	CNN, RF	Melanoma vs. nevi	AUC = 0.998 Accuracy = 98.2	CNN pre-trained models to predict diagnosis used in combination with a random-forest algorithm to classify WSIs as melanoma or nevi. Final predictions of the model based on validation datasets were compared to those of seven pathologists. The model was shown to outperform human pathologists.
Xie et al. [30] (2021)	841	H&E-stained images	CNN	Melanoma vs. nevi	AUROC = 0.962 Accuracy = 0.933	CNN model was trained using melanoma and nevi WSIs. A Grad-CAM method was then used to reveal the logic behind the CNN and understand the impact of specific areas in the model. Use of the Grad-CAM method and feature heatmaps revealed similarities between this group’s model and accepted pathological features.
Hekler et al. [31] (2019)	695	H&E-stained images	CNN	Melanoma vs. nevi	-	A pretrained model was adapted to a training dataset of melanoma and nevi histology images. This model was then tested on additional data and compared to diagnoses of a histopathologist. The misclassification rates for the trained model were 18% for melanomas and 20% for nevi.
Sankarapandian et al. [38] (2021)	7685	H&E-stained images	Deep learning system: hierarchical classification	Basaloid vs. squamous vs. melanoma high, intermediate, low risk and other	0.93 training 0.95 validation #1 0.82 validation #2	Hierarchical clustering was utilized for binary classification of nonmelanoma vs. melanocytic images followed by further classification of melanoma as “high risk” (melanoma) “intermediate risk” (melanoma in situ or severe dysplasia) or “rest” consisting of nonmelanoma skin cancers, nevus, or mild-to-moderate dysplasia.
Oskal et al. [40] (2019)	Train = 36 Test = 33	H&E-stained images	CNN	Epidermal segmentation	Mean positive predictive value = 0.89 ± 0.16 Sensitivity 0.92 ± 0.1	Epidermal regions were first annotated by an expert pathologist in Aperio ImageScope. WSIs were then split into small tiles, where a binary classifier categorized tiles as epidermis or non-epidermis.
Li et al. [33] (2021)	701	H&E-stained images	CNN	Melanoma vs. nevi	AUROC = 0.971 (CI: 0.952–0.990)	WSIs were divided into tiles, where individual predictions were made to assess the overall diagnostic probability of melanoma or nevi. Images diagnosed as melanoma were then further analyzed to determine the location of the lesion using a probability heatmap method.
Wang et al. [29] (2019)	155	H&E-stained images	CNN, RF	Melanoma vs. nevi	AUC = 0.998 Accuracy = 98.2% Sensitivity = 100% Specificity = 96.5%	Using whole-slide images of melanoma and nevi samples, regions of sample on the slide were segmented and split into tiles. The model was then used to create a tumor probabilistic heatmap and identify key features for melanoma diagnosis. Final binary classification of WSI was performed using a random-forest model based on extracted feature vectors. The performance of the model was compared to seven human pathologists, showing improved accuracy.
Sturm et al. [44] (2022)	99	H&E-stained images	CNN	Mitosis detection	Accuracy = 75%	Concordance of predeveloped algorithm along with multiple expert pathologists was tested for detection of mitoses in melanocytic lesions. Little improvement in diagnostic accuracy was observed with or without the aid of this algorithm (75% vs. 68%).
Ba et al. [32] (2021)	Train/Test = 781 Validation = 104	H&E-stained images	CNN, RF	Melanoma vs. nevi	Sensitivity = 100% Specificity = 94.7% AUROC = 0.99	The developed pipeline first extracts regions of interest then splits regions into tiles, where a CNN indicates whether the tile is a nevi or melanoma. Additional image features are then implemented into a random-forest classifier, where a final diagnosis for the WSI is made. The overall performance of the model matched that of the average dermatopathologist for differentiation between varying MPATH-DX categories.
Andres et al. [43] (2016)	59	H&E-stained images; Ki-67-stained slides	RF	Mitosis detection	83%	Mitoses were detected by assessing the individual pixel probability of contributing to mitotic cells. A significant correlation was found between the number of mitoses detected by their program and the number of Ki67-positive cells detected in Ki67-stained tissue slides.
Xie et al. [30] (2021)	2241	H&E-stained images	CNN	Melanoma vs. nevi	Sensitivity = 0.92 Specificity = 0.97 AUC = 0.99	Tested the efficacy of two different CNN for the differentiation of melanoma and nevi at four different magnification levels (4×–40×). The most accurate model was at 40× magnification.
Liu et al. [46] (2021)	227	H&E-stained images	CNN	Melanocytic proliferation segmentation	Accuracy = 0.927	A framework was developed for aiding pathologists in the identification and segmentation of melanocytic proliferations. Using sparse annotations generated by the pathologist, this pipeline finetunes the segmented regions using a CNN model on tile WSI regions.
Osborne et al. [49] (2011)	126	H&E-stained images	SVM	Melanoma vs. nevi	Sensitivity = 100% Specificity = 75% Accuracy = 90%	For differentiation between melanoma and nevi WSIs, this pipeline first removes irrelevant regions of the slide and then distinguishes the area of nuclei and cytoplasm in individual cells. Multiple features involving the nuclei are extracted including the number of nuclei per cell, nuclei-to-cytoplasm area ratio, perimeter, and shape.
Alheejawi et al. [45] (2020)	9	H&E-stained images	CNN, SVM	Melanocyte detection	Accuracy = 90%	CNN model segments individual nuclei within the image and extracts multiple morphological and textural features. SVM later classifies nuclei as normal or abnormal based on these features.
Zhang et al. [42] (2022)	30	H&E-stained images	CNN	Melanoma segmentation	Precision = 0.9740 Recall = 0.9861 Accuracy = 0.9553	WSIs of malignant melanoma were split into numerous tiles where developed CNN model differentiated patches, which represented benign and malignant tissues. The final model outputs a probabilistic heatmap of tumor regions and was found to outperform multiple similar predeveloped models for melanoma segmentation accuracy.
De Logu et al. [39] (2020)	100	H&E-stained images	CNN	Melanoma vs. healthy tissue patches	Accuracy = 96.5% Sensitivity = 95.7% Specificity = 97.7%	Based on WSIs of melanoma, slides were split into numerous tiles and individual tiles were classified using a pre-trained network as either melanoma or surrounding normal tissue.
Kucharski et al. [47] (2020)	70	H&E-stained images	DNN	Melanocyte nest detection	Dice similarity coefficient = 0.81 Sensitivity = 0.76 Specificity = 0.94	Based on a convolutional autoencoder neural-network architecture, a framework was implemented for the use of melanocytic nest detection. Slides were split into tiles, where individual tiles were classified as part or not part of a nest, eventually allowing for the segmentation of whole nests.
Kuiava et al. [37] (2020)	2732	H&E-stained images	CNN	Melanoma vs. basal vs. squamous vs. normal	Sensitivity = 92%, 91.6%, 98.3% Specificity = 97%, 95.4%, 98.8%	The efficacy of three different models to differentiate disease classifications was tested using a large dataset of melanoma, basal cell carcinoma, squamous cell carcinoma, and normal skin. All three models were able to differentiate between different types of cancer with high sensitivity and specificity.

### 3.2. Prognostic Applications

An accurate, individualized prognosis is essential for developing appropriate treatment and follow-up plans. Key melanoma prognostic factors include clinical, known histological, and molecular features; sentinel lymph node status; and radiologic imaging information about the distant and locoregional spread [7]. Time-tested histological prognostic features are some of the best predictors of outcome. They include the presence of ulceration, the presence and rate of mitoses and depth of invasion, and the Breslow thickness [6,8]. In addition, immunohistology features of melanoma and the overlying epidermis are also emerging as novel prognostic biomarkers along with gene-expression profiles of the tumor and along with markers of mutation burden and specific features of driver mutations which allow targeted melanoma therapy [7,8,50,51,52].

Digital histology images contain far more pixels than other commonly used medical imaging techniques, such as magnetic resonance imaging (MRI) and computerized tomography (CT) [18]. However, there are limited histological biomarkers large enough to be observed by the human eye. Deep learning offers a path to access this hidden wealth of information in digital histology images. Kulkarni et al. developed a deep neural network to predict whether a patient would develop distant metastasis recurrence [53]. This deep neural network uses a CNN to extract features followed by a recurrent neural network (RNN) to identify patterns, ultimately outputting a distant metastasis recurrence prediction. The models achieved AUCs of 0.905 and 0.88 when tested on validation datasets [53].

The sentinel lymph node status is considered a key prognostic factor of melanoma. However, this requires surgical excision of the first draining lymph node from the melanoma to provide a marker of overall nodal status. Despite being a strong prognostic indicator, the sentinel lymph node status used in combination with regional lymph node completion surgery has been found to have no benefit to disease-specific survival [54,55]. Brinker et al. developed an artificial neural network to predict the sentinel lymph node status based on H&E-stained slides of primary melanoma tumors. WSIs were split into tiles, where cell detection classified cells as tumor cells, immune cells, or others. After classification, cell features described in Kulkarni et al. [53] were extracted. Additionally, clinical features were implemented, including the tumor thickness, ulceration, and patient age. Image features were extracted with a pre-trained CNN model. The total slide classification was determined based on the majority classification of tiles. Clinical characteristics were also implemented into the model, including the tumor thickness, ulceration, and patient age. Overall, their most efficient model used a combination of image, clinical, and cell features and achieved an AUROC of 61.8% for classification between positive and negative sentinel lymph node status on the test dataset [56].

Targeted and immunotherapy of melanoma have revolutionized melanoma care. However, not all melanoma patients benefit from these therapies. Genomic testing of melanoma samples identifies tumors that will respond to targeted therapy, but immunotherapy response is harder to predict, and despite existing tools, novel markers for better patient selection for individualized therapy are needed [57,58]. Multiple models have been created to predict the immunotherapy response using melanoma histology image features [59,60]. Hu et al. predicted the progression-free survival based on WSIs derived from melanoma patients who received anti-PD-1 monoclonal antibody monotherapy [59]. Johannet et al. created a multivariable classifier to classify patients who received either anti-PD-1 or anti-CTLA-4 monotherapy as having a high or low risk of cancer progression [60]. This pipeline first used a segmentation classifier to distinguish between tumor, lymphocyte, and connective tissue slide tiles. They then implemented a response classifier to predict the response probability for each tile, ultimately leading to whole-slide classification based on the tile majority. Their final model achieved an ROC of 0.8–0.805 for the classification of progression-free survival after ICI treatment [60].

The presence and compositions of tumor-infiltrating lymphocytes (TILs), lymphocytic cells that have migrated to the tumor, correlate with the disease progression and response to immunotherapies [61]. The prognostic significance of TILs was initially somewhat controversial. Recent evidence suggests that the absence of TILs is a poor prognostic factor, while the brisk presence of TILs is associated with better disease-free survival [7,8]. There is also evidence that the quantity, localization, and phenotype of TILs are essential for predicting the response to immunotherapies and risk of disease progression [61]. Acs et al. developed an algorithm to recognize and segment TIL within WSIs and then calculate the frequency of these cells within each image [62]. Automated TIL scoring was found to be consistent with TIL scoring performed by a pathologist. Moore et al. then tested the ability of the automated TIL scores to predict patient outcomes [63]. Separating patients by those who did or did not die of melanoma found a significant correlation between the TIL score and disease-specific survival. To show the ability of their model to enhance the currently used methods of melanoma prognosis prediction, they tested the efficacy of their model to predict the prognosis in combination with patient information on the tumor depth and ulceration status. Overall, they found that the parameters discovered by their model contributed significantly to the overall prediction.

Studies published by Chou et al. further validated those found by Acs et al., using a TIL percentage score to predict overall survival outcomes [64]. Similar to previously described models, this model segmented regions of interest within the WSI, followed by the segmentation of various cell types, including TILs. Based on the well-known Clarke’s grading system of TIL scoring, they found little difference in the probability of recurrence-free survival. However, when using a newly defined low and high TIL score, they found significant differences between recurrence-free survival and overall survival probability. They, therefore, propose that this quantification of TIL may be more efficient for clinical use than the currently used methods. Based on this group’s model, Chou et al. further validated the ability of the model to interpret differences in recurrence-free and overall survival. Using a predeveloped neural network classifier that generates an automated TIL score in addition to human-based pathological analysis, this group sought to correlate automated TIL scoring with AJCC staging. They found that the percentage of TILs in the slide significantly improved the prediction of survival outcomes compared to Clarke’s grading. Using a threshold score of 16.6% TIL, they found significant differences in RFS (*p* = 0.00059) and OS (*p* = 0.0022) between “high”- and “low”-TIL-scoring patients [64].

BRAF mutations are common in melanoma [65]. Since the advent of targeted therapies, the BRAF mutation status provides essential clinical information [66]. Kim et al. initially trained an algorithm to define melanoma vs. nonmelanoma regions. Focusing on only regions of melanoma, they then tested the efficacy of three published BRAF mutation prediction classifiers. To better understand how BRAF mutated cells were distinguishable from the deep learning model, they performed pathomics analysis on these slides and found that cells with BRAF mutations showed larger and rounder nuclei [67]. In a later publication, they discovered that pixels located in the nuclei of cells were the most influential in predicting BRAF mutations. In their final prediction model, they combined clinical information, deep learning, and extracted nuclei features to predict BRAF mutation status in H&E WSIs of melanoma [68].

An accurate disease prognosis is essential for providing patients with an individualized treatment plan. New prognostic markers identified using deep learning show a significant advantage for use in combination with currently used markers such as ulceration and the Breslow thickness. Table 2 summarizes the prognostic applications of deep learning applications in melanoma histology.

## 4. Discussion

Accurate melanoma diagnosis and precise individualized prognostication of outcome are the cornerstones of appropriate melanoma management, crucial for driving therapeutic interventions and follow-up recommendations. Novel immunotherapies and targeted therapies are commonly used in metastatic melanoma management. Although these treatments have revolutionized melanoma therapy, caring for their numerous and sometimes fatal side effects has transformed oncology care over the past decade. There is currently a scarcity of tools to predict the treatment response and select appropriate therapy. The discovery of novel effective diagnostic and prognostic biomarkers in histological images could help revolutionize melanoma care and provide a more reliable workflow for making treatment-related decisions.

As the power and capability of computers increase, ML is becoming more frequently used for clinical decision-making. Although AI programs are not expected to replace physicians in the clinic any time soon, these models may soon provide secondary opinions by detecting information not seen by physicians and assisting in diagnosis and treatment-related decisions. Although further education will be required for both physicians and patients to understand the blackbox effect of ML better and gain trust in implementing these models in the clinic, AI-augmented medicine, as the next phase of the healthcare revolution, is already visible on the horizon.

Small datasets limit the ability of researchers to develop and test their ML models sufficiently. Large and heterogeneous datasets for training algorithms are expected to increase the efficacy of models and reduce the likelihood of overfitting. Additionally, robust validation is necessary to assess the accuracy of models. Methods such as cross-validation and split training/test datasets provide some insight into how a model may perform on new data and must be used in instances of limited accessible data. However, using an externally sourced validation dataset is a favorable approach as this will show that the model can be universally utilized. Given the need for a large amount of data for optimal training and validation, national or multinational consortia may aid the development of AI tools for melanoma diagnosis and prognostification.

Although several individual studies addressed the ability of deep-learning algorithms to finetune melanoma prognostic features in the past few years, only limited assessments of histological features have been performed in small- to medium-sized datasets. The ability of others to access published ML models on their own datasets will lead to increased reproducibility testing and validation of new ideas and foster high-quality team science. Therefore, it is also vital for researchers to make datasets and codes publicly available to allow the field of machine learning to grow for both histological and other data applications.

## 5. Conclusions

A wave of interest in personalized medicine has come, using individualized patient tumor and clinical information to identify optimal therapies for cancer patients. In recent years, AI has emerged as a powerful tool for identifying these individualized treatment plans. Robust and well-reviewed AI models are needed to drive these applications to the clinic. Personalized medicine has the potential to improve overall patient outcomes in cancer treatments. As new therapeutic options become available, treatment planning for patients becomes increasingly difficult. AI clinical models may help navigate these complex systems, using patterns that are unrecognizable to the eyes of physicians and scientists to provide patients with the most suitable therapeutic options.

## Figures and Tables

**Figure 1 cancers-14-06231-f001:**
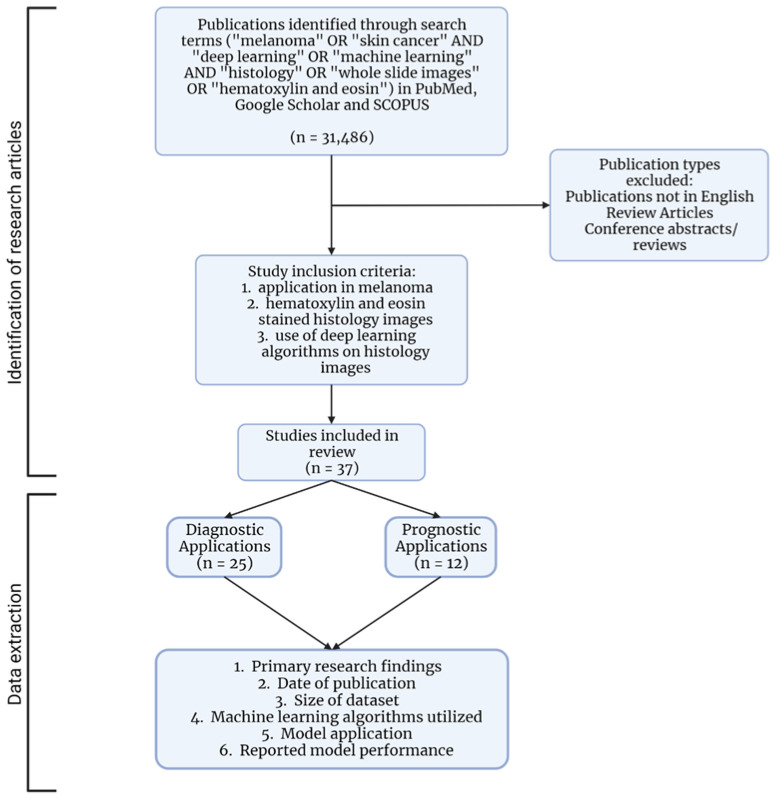
Overview of journal article identification and data-extraction methods.

**Figure 2 cancers-14-06231-f002:**
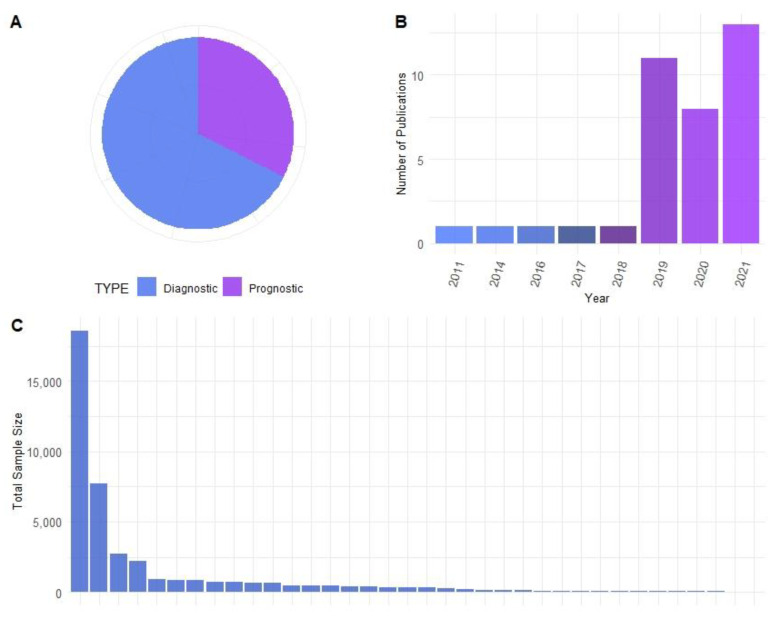
(**A**) Comparison of the number of publications with diagnostic or prognostic applications. (**B**) Number of papers published each year. As our literature search was conducted on 21 March 2022, years for this figure are defined by 21st March of the labelled year through 21st March of the following year. (**C**) Total size of dataset used for each study.

**Figure 3 cancers-14-06231-f003:**
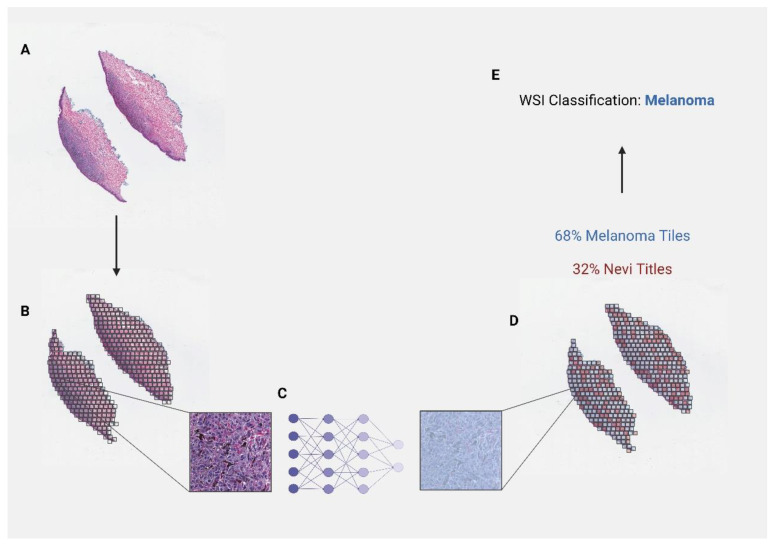
Schematic overview of image-processing workflow for deep-learning applications with whole-slide images. Example depicts application in diagnosis of nevi and melanoma. (**A**) Upload of digital whole slide image (**B**) Tiling of the image (**C**) Training of deep learning model using features extracted from individual tiles (**D**) Classification of individual tiles (**E**) Final image classification based on tile majority.

**Table 2 cancers-14-06231-t002:** Prognostic Applications.

Author	Dataset Size (# of Images)	Dataset Type	Model Type	Model Application	Reported Model Performance	Main Findings
Hu et al. [59] (2020)	476	H&E-stained images; clinical information	CNN	responder vs. non-responder	AUC of 0.778	The model predicted progression-free survival of melanoma patients that received anti-PD-1 monoclonal antibody monotherapy. Patients were classified as either responders or non-responders to therapy.
Kulkarni et al. [53] (2020)	Train = 108 Test #1 = 104 Test #2 = 51	H&E-stained images; clinical information	DNN	prediction of disease specific survival	AUC1 = 0.905 AUC2 = 0.88	Deep neural network predicted whether a patient would develop distant metastasis recurrence. The model used a CNN to extract features followed by an RNN to identify patterns, ultimately outputting a distant metastasis recurrence prediction.
Brinker et al. [56] (2021)	415	H&E-stained images; sentinel node status; clinical information	ANN	prediction of sentinel lymph node status	AUROC = 61.8%	The model predicted sentinel lymph node status based on WSI of primary melanoma tumors. Cell detection classified cells as tumor, immune, or other followed by further cell feature extraction. Additionally, clinical features were implemented including the tumor thickness, ulceration, and patient age.
Johannet et al. [60] (2021)	Train = 302 Validation = 39	H&E-stained images; clinical information	CNN	prediction of ICI response (PFS)	AUC = 0.8	Multivariable classifier to distinguish patients as having a high or low risk of cancer progression in response to ICI treatment. This pipeline first used a segmentation classifier to distinguish between different cell types then implemented a response classifier to predict the probability of response for each tile, ultimately leading to whole-slide classification based on tile majority.
Acs et al. [62] (2019)	641	H&E-stained images; clinical information	NN	TIL scoring	-	An algorithm recognized and segmented TIL within WSIs and then calculated the frequency of these cells within each image. Automated TIL scoring was found to be consistent with TIL scoring performed by a pathologist. It was also shown that automated TIL scoring correlated with patient prognostic factors.
Moore et al. [63] (2021)	Train = 80 Validation = 145	H&E-stained images; clinical information	NN	TIL scoring	-	By testing the ability of the automated TIL scores to predict patient outcomes, a significant correlation was found between the TIL score and disease-specific survival.
Chou et al. [64] (2021)	453	H&E-stained images; clinical information	NN	TIL scoring	-	Utilized a TIL percentage score to predict overall survival outcomes. Using a defined low and high TIL score, significant differences were found between the recurrence-free survival and overall survival probability.
Kim et al. [67] (2020)	Train = 256 Validation = 68	H&E-stained images; BRAF-mutant status	CNN	Prediction BRAF genotype	0.72 AUC test; 0.75 validation	Based on a multi-step deep-learning model and pathomics analysis to extract unique features of BRAF-mutated cells within melanoma lesions, it was found that cells with BRAF mutations showed larger and rounder nuclei.
Kim et al. [68] (2022)	Training = 256 Validation #1 = 21 Validation #2 = 28	H&E-stained images; BRAF-mutant status; clinical information	CNN	Prediction BRAF genotype	AUC = 0.89	After first extracting BRAF-mutated cell nuclei features, a final prediction model was created to combine clinical information, deep learning, and extracted nuclei features to predict the mutation status in WSIs of melanoma.
Forchhammer et al. [69] (2022)	831	H&E-stained images; clinical information	CNN	Overall survival	AUC = 0.694	The developed model was able to correctly classify low- and high-risk individuals based on the survival status during 2-year patient follow-up. There was a statistical difference in recurrence-free survival (*p* < 0.001) and AJCC Stage IV vs. AJCC Stage I-III (*p* < 0.05) between patients categorized in the low- and high-risk groups.
Phillips et al. [41] (2018)	50	H&E-stained images	CNN	Dermis vs. epidermis vs. tumor	-	The accuracy of segmentation of dermal, epidermal, and tumor regions by the model was compared to the segmentations of a pathologist. The multi-stride fully convolutional network proved to be diagnostically equivalent to pathologists, indicating a new automated resource for measurement of Breslow’s depth.
Chou et al. [64] (2020)	457	H&E-stained images; clinical information; pathologist generated TIL scoring	NN	Prediction of recurrence-free survival and overall survival		Using a predeveloped neural network classifier that generates an automated TIL score in addition to human-based pathological analysis, automated TIL scoring was correlated with AJCC staging. The percentage of TILs found in the slide significantly improved the prediction of survival outcomes compared to Clarke’s grading. Using a threshold score of 16.6% TIL, significant differences were found in RFS (*p* = 0.00059) and OS (*p* = 0.0022) between “high” and “low” TIL-scoring patients.

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
