# Peer review of "Diagnostic and Prognostic Deep Learning Applications for Histological Assessment of Cutaneous Melanoma"

_cancers, 2022, doi:10.3390/cancers14246231_

Round 1
Reviewer 1 Report
1. The Abstract should be more powerful and brief to discuss the following: the importance of the research, its objectives, the importance of artificial intelligence technology and the used methodologies.
2. The introduction is good, but in order to be complete it is necessary to add the shortcomings of manual diagnosis and the importance of artificial intelligence to treat lesions. The main contributions should be noted at the end of the introduction.
3. The types, locations "online" and data set sizes must be indicated.
4. The main section "2. Materials and Methods" contains ten lines and this is not acceptable. This section is the most important section in your research, where the materials, methods, techniques and algorithms are used. Subsections explaining the materials, methods, techniques and algorithms used in the research should be added.
5. A clear methodology must be drawn figure up to follow the work from input to output.
Author Response
Dear Reviewer #1,
We greatly appreciate your time and effort in reviewing our manuscript entitled “Diagnostic and prognostic deep learning applications for histological assessment of cutaneous melanoma”. Thank you for your thoughtful comments and all your recommendations to improve our manuscript. We followed all your recommendations. Please see below a detailed description of changes made to the manuscript to address each of your suggestions.
Comments and Suggestions for Authors
1. The Abstract should be more powerful and brief to discuss the following: the importance of the research, its objectives, the importance of artificial intelligence technology and the used methodologies.
The abstract has been rewritten to discuss the importance of the research, its objectives, the importance of artificial intelligence technology, and the methodologies used.
2. The introduction is good, but in order to be complete it is necessary to add the shortcomings of manual diagnosis and the importance of artificial intelligence to treat lesions. The main contributions should be noted at the end of the introduction.
Statements of the main contributions have been added to the introduction's second and fourth (last) paragraphs.
3. The types, locations "online" and data set sizes must be indicated.
Dataset type (H&E and/or IHC) and size are now indicated in Table 1.
4. The main section "2. Materials and Methods" contains ten lines and this is not acceptable. This section is the most important section in your research, where the materials, methods, techniques and algorithms are used. Subsections explaining the materials, methods, techniques and algorithms used in the research should be added.
The materials and methods section has been rewritten. Subsections describing data extraction of research articles and the creation of figures were added. Additionally, Figure 1 was added to describe the journal article identification and data extraction methods, and this also addresses comment #5.
5. A clear methodology must be drawn figure up to follow the work from input to output.
Figure 1 was added to describe the journal article identification and data extraction methods.
Thank you again for all your comments and suggestions.
Sincerely,
Gyorgy Paragh, MD, PhD
Reviewer 2 Report
The authors reviewed current deep learning applications of diagnostic and prognostic melanoma on histopathology. The manuscript was well structured. The content to be conveyed is communicated to the readers clearly. The manuscript is already good and does not seem to have much to improve upon. I have a suggestion. I would like to know a clear description of what type of data, HE stain. immunostain, or other image, was inputted in each study. I think this statement would be important in section 3.2 particularly, because prognostic states tend to be predicted by multi-omics data in the other fields.
Author Response
Dear Reviewer #2,
We greatly appreciate your time and effort in reviewing our manuscript entitled “Diagnostic and prognostic deep learning applications for histological assessment of cutaneous melanoma”. By addressing the concerns you brought up based on your expertise in the field, we have been able to greatly improve the quality of our manuscript. Below is a description of how what changes we made to the manuscript to address each concern.
Comments and Suggestions for Authors
The authors reviewed current deep learning applications of diagnostic and prognostic melanoma on histopathology. The manuscript was well structured. The content to be conveyed is communicated to the readers clearly. The manuscript is already good and does not seem to have much to improve upon. I have a suggestion. I would like to know a clear description of what type of data, HE stain. immunostain, or other image, was inputted in each study. I think this statement would be important in section 3.2 particularly, because prognostic states tend to be predicted by multi-omics data in the other fields.
Dataset types (H&E and/or IHC) are now indicated in Table 1.
Thank you again for your comments and suggestion.
Sincerely,
Gyorgy Paragh, MD, PhD
Reviewer 3 Report
The review “Diagnostic and prognostic deep learning applications for histological assessment of cutaneous melanoma” presents a really updated and comprehensive report on the relevant studies on artificial intelligence analysis application for the diagnosis and the prognosis of melanoma based on pathology image. The growing importance of the support of artificial intelligence-based analysis to histopathology is supported to help personalized medicine to assess optimal therapies for cancer patients. The review is acceptable as it is for publication.
Author Response
Dear Reviewer #3,
We greatly appreciate your time and effort in reviewing our manuscript entitled “Diagnostic and prognostic deep learning applications for histological assessment of cutaneous melanoma”.
Comments and Suggestions for Authors
The review “Diagnostic and prognostic deep learning applications for histological assessment of cutaneous melanoma” presents a really updated and comprehensive report on the relevant studies on artificial intelligence analysis application for the diagnosis and the prognosis of melanoma based on pathology image. The growing importance of the support of artificial intelligence-based analysis to histopathology is supported to help personalized medicine to assess optimal therapies for cancer patients. The review is acceptable as it is for publication.
We made all modifications recommended by the reviewers. Thank you again for your comments and for reviewing our manuscript.
Sincerely,
Gyorgy Paragh, MD, PhD
Round 2
Reviewer 1 Report
The authors answered my questions